# Comparison of Drying Methods and Their Effect on the Stability of Graševina Grape Pomace Biologically Active Compounds

**DOI:** 10.3390/foods11010112

**Published:** 2022-01-01

**Authors:** Tea Sokač, Veronika Gunjević, Anita Pušek, Ana Jurinjak Tušek, Filip Dujmić, Mladen Brnčić, Karin Kovačević Ganić, Tamara Jakovljević, Darko Uher, Grozdana Mitrić, Ivana Radojčić Redovniković

**Affiliations:** 1Faculty of Food Technology and Biotechnology, University of Zagreb, Pierotti St. 6, 10000 Zagreb, Croatia; tsokac@pbf.hr (T.S.); vgunjevic@pbf.hr (V.G.); apusek@pbf.hr (A.P.); ana.tusek.jurinjak@pbf.unizg.hr (A.J.T.); filip.dujmic@pbf.unizg.hr (F.D.); mbrncic@pbf.hr (M.B.); kkova@pbf.hr (K.K.G.); 2Croatian Forest Research Institute, Cvjetno Naselje 41, 10450 Jastrebarsko, Croatia; tamaraj@sumins.hr; 3Faculty of Agriculture, University of Zagreb, Svetošimunska 25, 10000 Zagreb, Croatia; duher@agr.hr; 4Kutjevo d.d., Kralja Tomislava 1, 34340 Kutjevo, Croatia; grozdana.mitric@kutjevo.com

**Keywords:** Peleg model, biologically active compounds, polyphenols, tannins, tartaric acid, grape pomace, vacuum drying, open sun drying, hot air drying

## Abstract

Valorisation of grape pomace, a by-product of the winery industry, has been pushed into the spotlight in recent years since it can enable lower environmental impact, but it can also bring an added value to the wine production process by recovering several grape pomace biologically active compounds. The first step that allows for grape pomace reuse is its drying, which should be carefully performed in order to preserve the biologically active compounds’ stability. In this study, the effects of different drying methods on the stability of polyphenols, tannins and tartaric acid in grape pomace (*Vitis vinifera*) cv. Graševina were investigated. In particular, vacuum drying (at different temperatures: 35, 50 and 70 °C), conventional drying at 70 °C and open sun drying were performed and the drying kinetics was described using Peleg’s model. Considering the processing time and thermodynamics, vacuum drying at 70 °C was the most convenient processing method. Polyphenols were highly stable during drying, and slight degradation occurred during vacuum drying at 35 and 50 °C. Tannins and tartaric acid were more prone to degradation depending on the drying method applied and showed the greatest stability during vacuum drying at 70 °C.

## 1. Introduction

Grape pomace is the most abundant biowaste of the wine industry, which roughly consists of grape stalks, seeds and skins [1]. It is common knowledge that wine production is a significant agricultural sector in the Republic of Croatia. The total production of grapes in the 2020 wine year was 125.043 tons, which generally means making large amounts of solid waste after wine production [2]. According to the FAO/WHO, the world’s wine industry produces 5–10 million tons of waste in one year, while for Croatia, this means an annual production of more than 15,000 tons of solid waste, where the transformation or disposal of this waste is problematic in both ecological and economic terms [3,4,5]. The leading Croatian white grape variety, namely, Graševina, accounts for more than 60% of the total wine area of continental Croatia. Hence, due to the wide distribution of Graševina, the largest share of wine waste is made by processing this variety.

As a result of the increased awareness of the need to reduce the environmental impact of organic waste and the attention toward the sustainability of agricultural practices, efforts have been made to use grape pomace in different fields of industry [6]. Accordingly, most interest has been paid to the recovery of biologically active compounds from grape pomace, which is a rich source of mainly phenolic compounds that are associated with significant antioxidant activity [7]. Phenolics not only include molecules with one phenol ring, such as phenolic acids and phenolic alcohols, but also a plentiful variety of molecules with a polyphenol structure (i.e., several hydroxyl groups on aromatic rings). Due to their structural diversity, there is a wide range of phenolic compounds that occur in nature [8]. Many phenolics were identified in grape pomace, where the most abundant can in general be classified into three main groups as phenolic acid (mainly benzoic and hydroxycinnamic acids), simple flavonoids (catechin and flavonols) and tannins [9,10]. Phenolic compounds were reported to have multiple biological activities, including cardio-protective, anti-inflammatory, anti-carcinogenic and antibacterial properties attributed mainly to their antioxidant and antiradical activity [11,12]. Furthermore, tartaric acid is another interesting metabolite recovered from grape pomace, that finds many applications as an acidification agent and taste enhancer alongside its antioxidant, pH regulatory and preservative activities, in the food, bakery and pharmaceutical industries [13,14]. Due to the high content of biologically active compounds present in grape pomace, considerable effort has been employed to optimize the isolation of these compounds and expand their use in the food, pharmaceutical and cosmetic industries [15,16,17]. In this regard, drying can be recommended as an effective method for grape pomace processing, resulting in a product that is plentiful in biologically active compounds that can be used as an ingredient or raw material in the formulation of other food products [18,19]. Moreover, the drying of fresh plant material has long been used to acquire stable products that can be stored for a prolonged time.

There is a direct connection between the drying procedures and the fundamental physical compliance rules of mass and heat transfer in capillary porous materials. Furthermore, the basic mechanism of mass and heat transfer is complex; thus, the choice of the proper drying procedure is of huge importance that involves many aspects, such as the overall drying conditions, nature of the material and initial moisture content [20]. Due to the mentioned drying advantages, the dehydration of wet grape pomace becomes an essential process prior to any further application [21,22]. There are many different drying methods that have been applied to grape pomace, from the most basic technique, such as solar/sun drying, to more expensive methods, such as freeze or microwave drying. In recent years, to produce dried foods with higher nutritional and sensorial attributes, nonconventional drying methods have been employed, such as ultrasound, pulsed electric field, high pressure and combinations of drying methods [23]; however, initial investments in these methods are very high, which is the main reason why they are rarely used in large-scale industrial implementations.

Throughout drying, the temperature is the key determining variable for maintaining biologically active components [19]. Temperature is directly related to mass and heat transfer and, in the case of thermosensitive compounds, can be responsible for the preservation or degradation of biologically active compounds. Polyphenolics are heat and oxygen sensitive; therefore, this demands special attention. Vacuum drying is a process in which materials are dried in a reduced pressure environment, which lowers the heat needed for rapid drying and thereby reduces the possibility of the thermal decomposition of the product. Several studies have evaluated the effects of different drying methods on the biochemical changes of grape pomace [24,25,26]. However, the degradation of biologically active compounds separately in white grape seeds, skins and pomace throughout the drying process has been rarely investigated and reported in papers. This research could be useful to detect changes in the composition of biologically active compounds during drying, as well as optimize the drying processes as a pre-treatment in the production of value-added products.

However, to fully utilize grape pomace, it is critical to define drying conditions that can maximize the retention of biologically active compounds while remaining economically feasible on a larger industrial scale. Thus, the aim of this study was to investigate the stability of biologically active compounds in Graševina grape pomace, seeds and skins under different drying conditions, including 35 °C, 50 °C and 70 °C vacuum drying, 70 °C conventional drying and open sun drying. In addition, the experimental data were fitted to Peleg’s model in order to estimate the parameters of drying kinetics and to predict the equilibrium moisture content.

## 2. Materials and Methods

### 2.1. Chemicals and Materials

All chemicals and standards were purchased from Sigma Chemical Co. (St. Louis, MO, USA). All solvents were of HPLC grade. *Vitis vinifera* cv. Graševina grape pomace was obtained from the Croatian native grape cultivar Kutjevo d.d. in September 2020.

### 2.2. Sample Preparation

The organic waste from wine production was purified from residual waste, leaves and parts of stalks. Samples of Graševina grape seeds, skins and pomace were used for the analyses. Prior to conducting the experiments, the amount of total dry matter and residual moisture of Graševina grape pomace, seeds and skins samples were determined according to the Association of Official Analytical Chemists [27] official methods of analysis using a conventional dryer (Instrumentaria ST-05 Sterilizer, Zagreb, Croatia) at 103 ± 2 °C under pressure ≤ 100 mm Hg (13.3 kPa).

### 2.3. Drying Methods

Graševina grape pomace, seed and skin samples were dried by applying three different methods as follows. Vacuum drying was conducted by applying a vacuum dryer (Memmert GmbH + Co. KG, Schwabach, Germany) set at 35, 50 and 70 °C at 100 mbar. Hot air drying was carried out in a conventional dryer using natural convection at room pressure and a fixed temperature of 70 °C. Due to the significant amounts of residual grape pomace in the industry, open sun drying as a traditional and energy-free method was performed outside under direct sunlight and atmospheric pressure. During dehydration in the open sun drying system, the weight loss, ambient air temperatures and relative humidity were measured using a digital thermo-hygrometer every hour from 8:30 a.m. to 4:30 p.m. The sun-dried samples were kept in a closed desiccator overnight. The desiccator without hygroscopic salt was used as a container that preserved the biomass moisture. Detailed drying conditions for every process are reported in Table 1.

In general, moisture losses of the samples were recorded at different specific intervals depending on the drying methods. The weights of the samples collected during each drying test were converted into moisture contents on a dry basis. Drying was completed when the final seed moisture content was around 5–8% due to the requirement of potential further processing and in pomace and skin when the weight was constant for several consecutive measurements. The results of the drying kinetics for the three methods (vacuum, conventional and open sun drying) were expressed as a function of the moisture content and reported in percentages (%). The experimental data were adjusted using Peleg’s model.

### 2.4. Drying Kinetic and Thermodynamic Considerations

Peleg [28] proposed an empirical equation to describe the sorption characteristics of various food materials. Using short-time experimental data for predicting the equilibrium moisture content of foods and grains is the major advantage of the model. The model for the drying process is shown as [28]:(1)$$M_{t}=M_{0}-\frac{t}{K_{1}+K_{2}t}$$
where *M_t_* is the moisture content expressed as dry matter at time *t*, *M*_0_ is the initial moisture content expressed as dry matter, *t* is the time (h), *K*_1_ is the Peleg rate constant (h %^−1^) and *K*_2_ is the Peleg capacity constant (%^−1^).

The Peleg rate constant *K*_1_ relates to the desorption rate at the beginning (*R*_0_), i.e., *R* at *t* = *t*_0_:(2)$$R=-\frac{1}{K_{1}}$$

Linearization of Equation (1) gives:(3)$$\frac{t}{M_{t}-M_{0}}=K_{1}-K_{2}t$$

The plotting of Equation (3) is a straight line where the first term of the second member is the intercept (*K*_1_) and *K*_2_ is a slope. The major advantage of the Peleg model is to save time by predicting the water sorption kinetics of foods, including the equilibrium moisture content [29]. The parameters of Peleg’s model were estimated by fitting the mathematical model to the experimental data by applying Statistica software, version 8 (Statsoft Inc., Tulsa, OK, USA). Shafaei et al. [30] indicated that *K*_1_ could be compared to a diffusion coefficient and the Arrhenius equation could be used to describe the temperature dependence of the reciprocal of Peleg’s constant *K*_1_ in the following manner:(4)$$\frac{1}{K_{1}}=K_{\mathrm{ref}}\exp\left[-\frac{E_{a}}{R}\left(\frac{1}{T}-\frac{1}{T_{\mathrm{ref}}}\right)\right]$$
where *K*_1_ is the Peleg rate constant (h g g^−1^), *K*_ref_ is the frequency factor (h^−1^); *E_a_* is the activation energy (kJ mol^−1^); *R* is the universal gas constant (8.314 J mol^−1^ K^−1^); and *T* and *T*_ref_ are the drying temperature and reference temperature (K), respectively. To reduce the co-linearity of *K*_ref_ and the activation energy, the reference temperature was selected as the average temperature of the experiment.

After linearization, Equation (4) becomes [30,31]:(5)$$\ln\left(\frac{1}{K_{1}}\right)=lnK_{\mathrm{ref}}+\left(\frac{E_{a}}{R}\right)\left(\frac{1}{T_{\mathrm{ref}}}-\frac{1}{T}\right)$$

If ln(1/*K*_1_) is plotted against ((1/*T*_ref_) − (1/*T*)), a straight line with a slope (*E_a_*/*R*) is obtained, from which the activation energy can be calculated and the constant (*K*_ref_) can be assessed.

Furthermore, the *E_a_* value allows for the determination of different thermodynamic parameters, such as the enthalpy (Δ*H*), the entropy (Δ*S*) and the free energy (Δ*G*) using the following equations [30]:(6)$$\Delta H=E_{a}-R$$
(7)$$\Delta S=R\left(\ln A-\ln\frac{k_{B}}{h_{p}}-\ln T\right)$$
(8)ΔG=ΔH−TΔS
where ln*A* is the ordinate intersection when regression analysis is applied to the plot obtained in the calculation of *E_a_*, *k_B_* is Boltzmann’s constant (1.38 × 10^−23^ J K^−1^), *h_p_* is Planck’s constant (6.626 × 10^−34^ J s) and *T* is the absolute temperature.

### 2.5. Preparation of Extracts for Determination of Biologically Active Compounds

In order to evaluate the effect of drying on the biologically active compounds of the Graševina grape pomace, the contents of the total phenolic compounds, tannins and tartaric acid of the samples taken at predetermined times throughout the drying process were evaluated.

Samples of grape pomace, skins and seeds sampled during drying were ground and immediately extracted according to the method reported by Carmona-Jiménez et al. [32] and Palma and Barroso [33] with some modification. The extraction procedure was carried in an ultrasonic bath (XUB Series Digital Ultrasonic Baths, BioSan, Riga, Latvia) for 30 min at 40 °C. Solid–liquid ratios of 0.06 g per mL of aqueous ethanol (70% of ethanol), were used for extraction. The extracts were filtered and the supernatant was adjusted to a final volume of 10 mL (0.05 mg mL^−1^) and stored at +4 °C for one day due to the large numbers of extractions that were performed for every drying process observed. The obtained extracts were then analyzed the following day for total polyphenols, tannins and tartaric acid contents.

#### 2.5.1. Total Polyphenols Content (TPC) Determination

Total phenolic content (TPC) was determined using the Folin–Ciocalteu method [34]. The absorbance was measured at 760 nm (Specord 50 PLUS UV/VIS spectrophotometer, Analytik Jena, Jena, Germany), and the results were expressed as milligrams of gallic acid equivalent per gram of dry matter (mg GAE g^−1^_dm_). All analyses were performed in triplicate.

#### 2.5.2. Total Tannins Content (TTC) Determination

The total tannins content (TTC) was determined using the Bate-Smith method as described in [35]. The proanthocyanidin concentration was obtained by multiplying the difference in absorbance at 550 nm (Specord 50 PLUS UV/VIS spectrophotometer, Analytik Jena, Jena, Germany) between tube A_2_ and tube A_1_ by 19.33, which is the absorptivity coefficient of cyanidin after the acidic cleavage of the condensed tannins (Bate-Smith reaction), as summarized in Equation (9): (9)$$Total\;tannins\;\left[{g\;L}^{-1}\right]=19.33\cdot \left(A_{1}-A_{2}\right)$$
where *A*_1_ is the absorbance of the hydrolyzed sample, *A*_2_ is the absorbance of the unhydrolyzed sample and 19.33 is the calculation factor. The TTC is expressed as the tannins mass over the mass of dry matter (mg g^−1^). All analyses were performed in duplicate.

#### 2.5.3. Tartaric Acid Analysis

Tartaric acid was quantified using an HPLC system (1260 Infinity II, Agilent, Santa Clara, CA, USA) coupled with a diode array detector (UV/DAD, 1260 Infinity II, Agilent, USA) and an automatic sampler (1260 Infinity II, Agilent, Santa Clara, CA, USA). Separation was achieved using isocratic elution water at pH 2.5 in a Poroshell 120 SB C18 column (150 mm, 4.6 mm, 5 µm; Agilent, Santa Clara, CA, USA). The flow rate was 1 mL min^−1^ and the elution was performed for 7 min. The sample injection volume was 15 µL and the samples were always filtered through 0.22 µm polytetrafluoroethylene (PTFE) filters prior to injection. UV-DAD acquisitions were carried out in the 200–600 nm range, while the chromatogram was acquired at 210 nm. Tartaric acid identification was done by comparing the retention time and UV spectrum with a tartaric acid external standard. The quantification was performed by considering the standard calibration curve prepared in water at pH 3 acidified with sulphuric acid (five points from 0.1 to 1 mg mL^−1^). All analyses were performed in triplicate and results are expressed as the average.

### 2.6. Statistical Analysis

Statistical analyses were performed using Statistica software, version 10 (Statsoft Inc., Tulsa, OK, USA). Differences between the means of the polyphenols, tannins and tartaric acid content results, as well as the results of the moisture content monitoring during drying, were tested using analysis of variance (ANOVA) at the significance level of *p* < 0.05, followed by Tukey’s HSD test. The applicability of the drying model was estimated based on the coefficient of determination (*R*^2^) and root mean square error (RMSE). The coefficient of determination *R*^2^ represents the proportion of the variance in the dependent variable, which is explained by the linear regression model (Equation (10)). RMSE measures the standard deviation of the residuals (Equation (11)).
(10)$$\;R^{2}=1-\frac{\sum^{\;}{\left(y_{i}-\hat{y}\right)}^{2}}{\sum^{\;}{\left(y_{i}-\bar{y}\right)}^{2}}$$
(11)$$RMSE=\sqrt{MSE=}\sqrt{\frac{1}{N}\overset{N}{\underset{i=1}{\sum }}{\left(y_{i}-\hat{y}\right)}^{2}}$$

## 3. Results and Discussion

### 3.1. Kinetics and Thermodynamics of Grape Seeds, Skins and Pomace Drying

In this work, three different drying methods (drying in a vacuum dryer, conventional drying and open sun drying) of grape seeds, skins and pomace were carried out and compared in order to find the most suitable method for drying considering the processing time, needed energy and necessary equipment. The initial moisture content for the grape seeds was approximately 28.96 ± 4.21%, for skins 63.58 ± 3.29% and for grape pomace 49.36 ± 3.45%.

The drying process in a vacuum dryer was carried out at three different temperatures. For materials sensitive to thermal damage, a vacuum dryer may be used to reduce the drying temperature and pressure to protect the grape pomace physico-chemical characteristics. Figure 1 shows the effect of different temperatures on the drying kinetics of the grape pomace. As expected, the moisture content decreased during the drying processes. The highest temperature resulted in the shortest drying time. The equilibrium moisture content was reached at different process times. After drying for 12 h at 35 °C, 5 h at 50 °C and 3 h at 70 °C, the equilibrium moisture content for seeds was around 4.64 ± 0.84%, for skins was around 6.99 ± 0.83% and for pomace was around 8.32 ± 0.96%. Furthermore, the initial moisture content was the highest for grape skins and it dried up the most, followed by grape pomace and seeds.

As mentioned before, the drying kinetics was described using Peleg’s model. The model has shown good applicability to dehydration and rehydration processes and it was applied to various food materials by other researchers and was found to be suitable [36,37,38].

The constants of Peleg’s model obtained from the mathematical modeling at different drying temperatures are shown in Table 2.

Table 2 shows the constants of Peleg’s equation (*K*_1_, *K*_2_), coefficient of determination (*R*^2^) and rate of desorption (*R*_0_) for different drying methods for grape seeds, skins and pomace. The constant *K*_1_ is related to the mass transfer rate, where the higher the *K*_1_, the lower the initial water desorption rate. The constant decreased as the temperature increased, suggesting an increase in water desorption rate. The second constant *K*_2_ is related to the maximum water absorption capacity, where the lower the *K*_2_, the higher the water absorption capacity [39]. It is noted from Table 2 that there was a small decrease in values of constant *K*_2_ when the temperature increased. Researchers also reported the decrease of both constants with temperature [38,40]. The obtained coefficients were above 0.95, which suggests that the Peleg equation fit the experimental data and was suitable for describing the decrease in moisture content in grape pomace [41]. As can be noticed from all the results, the rate of desorption was the highest for the drying process in a vacuum dryer at 70 °C, which means that the drying of grape seeds, skins and pomace was the fastest at this temperature. Using a higher temperature resulted in a higher value of the rate of desorption. Comparing the rate of desorption for seeds, skins and pomace, it can be noticed that the lowest rate was for seeds, then skins and the greater values were obtained for grape pomace.

Moreover, Peleg’s constant (*K*_1_) can be used for calculating the activation energy and thermodynamic parameters, such as differential enthalpy, entropy and Gibbs energy (Table 3 and Table 4). These thermodynamic parameters characterize the drying process.

Enthalpy is a thermodynamic property of a system and it is related to the energy needed to remove water bound to the product during the drying process. The differential enthalpy decreased with increasing temperature. At lower temperatures, the values of differential enthalpy were higher, which indicated that a greater amount of energy was required to promote drying. The values were positive for grape seeds, skins and pomace, which indicated that the process was endothermic, in other words, heat should be brought to the system. Similar results were reported in a study by Correa et al. [40]. Entropy is a thermodynamic property that can be associated with the level of disorder between water and the product. In the drying process, the entropy became more negative as the temperature of drying increased. A negative change in entropy indicates that the disorder of an isolated system has decreased. It was noticed that there was a small difference in entropy values for a temperature difference of 20 °C, which can be explained by the theory of activated complexes in which a substance in an activation condition may acquire negative entropy if the degrees of freedom of translation or rotation are lost during the formation of the activated complex [42]. The Gibbs free energy increased when the temperature increased. For all materials dried in a vacuum dryer, the values for the Gibbs energy were positive, which indicated that the process was not spontaneous. In principle, a drying process requires the addition of energy from the environment to promote a reduction in the moisture content [40].

Comparing thermodynamic parameters for the grape seeds, skins and pomace at the determined temperatures, the highest values for enthalpy, entropy and Gibbs energy were obtained for grape seeds, followed by grape pomace and skins. This can be related to the structure of grape seeds, which have greater resistance to water loss and requires more energy for the drying process [43]. Furthermore, for the same reason, the activation energy was the highest for seeds (Table 3).

The vacuum drying method was found to be an effective method for drying in industries due to the conservation of energy, i.e., less energy is needed for drying. On the other hand, this method requires additional equipment, which includes higher operational costs and control of the process is complicated due to maintaining the lower pressure [44]. Considering the lower operational costs, another method for grape pomace drying was conventional drying at 70 °C because this temperature was the most effective for vacuum drying. In a conventional dryer, the heat is transferred via conduction and it took 7 h to achieve the equilibrium moisture content (Figure 1) and the content was around 5–8% for all materials, which was the same as that obtained for vacuum drying.

Due to the huge amount of obtained grape pomace in a few weeks during the season of grape ripening, the mentioned methods could cause a bottleneck due to equipment capacity. Open sun drying can be an alternative method because it does not require any special equipment and consumed energy but it should be considered that the conditions of air temperature and humidity are not constant; therefore, it may take a much longer time. Thus, the open sun drying method was also carried out in this work. The average temperature during the open sun drying was 31.99 °C and the humidity was 40.57%. This method was carried out for 26 h to achieve the equilibrium moisture content and it was approximately the same as for the previous methods (Figure 1).

The Peleg’s constants, rate of desorption and equilibrium moisture content for the drying in a conventional dryer and open sun drying are shown in Table 2. Comparing the drying in a vacuum dryer and in a conventional dryer at 70 °C, it can be noticed that the rate of desorption had higher values for drying in a vacuum dryer. On the other hand, the desorption rate was the lowest for open sun drying.

Considering all the results obtained for the different drying methods, the most suitable method for grape pomace was drying in a vacuum dryer at 70 °C. In terms of the necessary time for the drying process and thermodynamic parameters, this method was acceptable. The calculated parameters demonstrated the considerations relating to the drying process.

### 3.2. Biologically Active Compounds’ Stability

Preserving the stability of grape pomace phytochemicals is essential for its further use. It is well known that biologically active plant compounds are quite unstable under elevated temperatures [45]. Therefore, in this study, close attention was paid to the most significant biologically active grape pomace compounds during different drying processes, in particular, polyphenols, tannins and tartaric acid. Due to their biological activities, polyphenols and tannins are high-value ingredients for functional foods, beverages, cosmetics and pharmaceutical formulations [15], while tartaric acid is likewise of great interest in the food, beverage, cosmetic and pharmaceutical industries [26,46,47]. Given the various possibilities of polyphenols, tannins and tartaric acid applications, their preservation is crucial to exploit their natural bioactivities and to reuse grape pomace as a valuable by-product. To assess the biologically active compounds’ stability, it was necessary to isolate the aforesaid compounds from the grape pomace. Solid/liquid extractions were performed during drying and the extracts were analyzed. The results are presented in Table 5.

The starting polyphenols content was the highest in grape seeds (66.23 ± 3.97 mg g^−1^), followed by pomace (29.17 ± 3.54 mg g^−1^) and skins (18.67 ± 3.81 mg g^−1^), which is in agreement with previous studies indicating that white grapes seeds are much richer in polyphenols than the rest of the grape fruit [48,49]. In general, Graševina grape skins, seeds and pomace polyphenols were quite stable during any type of drying (Table 5). Teles et al. [25] monitored red grape pomace polyphenols’ stability in a furnace muffler at 40, 50 and 60 °C. The polyphenols stability was the highest at 60 °C. This drying was also the shortest. The authors suggested that the higher temperatures deactivated phenol dehydrogenases, diminishing the possibility of enzymatic polyphenols degradation. In a study by Lopez-Vidana et al. [50], the authors monitored Jaboticaba barry polyphenols stability when performing drying in a conventional dryer at 40, 50 and 60 °C. The authors likewise found that the polyphenols stability was preserved at 50 and 60 °C, while degradation occurred at 40 °C. In a similar study, grape pomace drying in a furnace muffler at 60 °C was monitored. Likewise, the polyphenols degradation did not occur. The degradation also did not occur when the biomass was air dried, even though this drying was performed over an extended period [51]. Flavanoids are generally the most abundant polyphenols family in grape pomace and their stability was monitored in Cabernet and Muscadine grape pomace under different drying conditions in a study by Yu [52]. During drying at room temperature for 7 days and drying at 70 °C in a vacuum dryer for 24 h, flavanoids were stable in Cabernet grape pomace. However, flavonoids degradation in Muscadine grape pomace was seen in both drying scenarios. Therefore, the polyphenols degradation also highly depends on the grape varieties. Generally, flavonoids are quite stable under high temperatures [53]. Carmona-Jiménez et al. [32] performed drying of five red grape pomace varieties in a climatic chamber at 40 °C. During the drying process, polyphenols deterioration was not observed. Instead, the authors found that the drying process even enhanced the grape pomace polyphenols’ extractability. In particular, drying causes cell wall breakage and destruction and, consequently, allows for easier extraction [54]. This effect was not observed in our study. The results in Table 5 also show that the polyphenols’ stability did not depend on the type of dryer used. When comparing drying in a vacuum and conventional dryer performed at the same temperature (70 °C), the polyphenols stability was preserved in both cases. Degradation caused by oxidation was expected when performing drying in a conventional dryer due to the higher oxygen amount [45].

The stability of tannins in grape pomace, skins and seeds was best preserved when drying was performed in the vacuum dryer at 70 °C. On the other hand, the extent of tannins degradation was highest in the conventional dryer at 70 °C. The latter is not surprising since tannins are unstable under high temperatures, light and the presence of oxygen [55]. In a study, red grape pomace tannins’ stability was assessed while drying in a furnace muffler. When drying grape pomace at 60 °C, tannin degradation occurred. At 40 °C, there was no degradation and the tannins’ stability was preserved for 3 days at this temperature [52]. However, when drying biomass in our study at 35 °C in the vacuum dryer, slight degradation was observed. Open sun drying likewise caused mild tannin degradation. This could have been due to the activity of polyphenols oxidase, which is generally stable at temperatures ranging from 25 to 65 °C [56]. However, the initial tannins content was highest in grape seeds, namely 73.88 ± 12.92 mg g^−1^. The initial tannins content in grape pomace and skins was 41.54 ± 8.03 mg g^−1^ and 29.45 ± 3.84 mg g^−1^, respectively. These results indicated that a higher tannins content can be found in grape seeds. Similar results have already been reported in the literature [57,58].

The initial tartaric acid content was the highest in grape skins (5.16 ± 0.58 mg g^−1^), while it was the lowest in grape seeds (0.46 ± 0.12 mg g^−1^). In pomace, the starting tartaric acid content was 3.54 ± 0.28 mg g^−1^. Tartaric acid degradation was the lowest when drying the grape pomace, skins and seeds in a vacuum dryer at 50 and 70 °C. The degradation was the highest when the grape pomace, skins and seeds were dried in a conventional dryer at 70 °C. Rösti et al. [59] dried Merlot and Shiraz grapes berries at 9, 15, 21 and 27 °C. The tartaric acid degradation in Merlot and Shiraz was 35.5% and 48.5%, respectively. The authors presumed that due to the potassium concentration increase, tartaric acid precipitated as potassium hydrogen tartrate, causing loss during the sampling. Tartaric acid stability at room temperature and atmospheric pressure was monitored in a study by Clark et al. [60]. Tartaric acid solution was prepared and its stability was monitored over 10 days in outdoor conditions during summer in Australia. The solution was shown to be unstable, especially in the presence of light. Moreover, it was shown that the iron content in the solution has a substantial effect on tartaric acid oxidation. Tartaric acid degradation also occurred when the grape pomace was stored in outdoor conditions, but the authors reported it to be a result of unwanted pomace fermentation [26].

Overall, drying at 70 °C provided the highest stability of all the monitored compounds. Furthermore, this drying process was the fastest, therefore offering time and energy savings.

## 4. Conclusions

This study was dedicated to finding the most suitable method for grape pomace, seeds and skins drying that will ensure a fast process while preserving the biologically active compounds against deterioration. Obtained results showed that drying in a vacuum dryer at 70 °C provided the fastest drying, saving time and energy. The results also demonstrated that drying grape pomace would be more energy efficient on an industrial level than drying it separately as grape skins and seeds. Considering the biologically active compounds, polyphenols were stable during all drying methods tested. Namely, only a slight polyphenol deterioration occurred when drying at 35 and 50 °C. On the other hand, tannins were extremely unstable during the conventional and open sun drying processes. Nevertheless, drying at 70 °C in the vacuum dryer led to tannins degradation to a lesser extent. Tartaric acid was also shown to be prone to degradation, particularly when performing conventional drying at 70 °C and open sun drying. Similarly to the tannins, the tartaric acid’s stability was preserved when the drying was carried out in the vacuum dryer at 70 °C. No significant differences between the stability of the biologically active components of grape pomace and separated skins and seeds were observed. This study provides valuable information on grape pomace pretreatment for the realization of grape pomace valorization.

## Figures and Tables

**Figure 1 foods-11-00112-f001:**
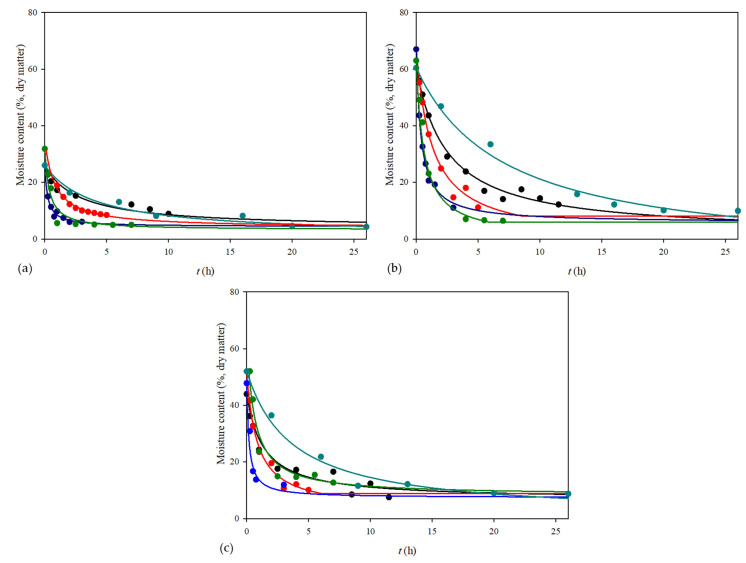
Drying curves obtained using different drying methods for (**a**) grape seeds, (**b**) grape skins and (**c**) grape pomace (• vacuum dryer at 35 °C, • vacuum dryer at 50 °C, • vacuum dryer at 70 °C, • conventional dryer at 70 °C, • open sun drying—Peleg’s model).

**Table 1 foods-11-00112-t001:** Drying conditions for the performed drying processes.

Drying Process	Temperature (°C)	Pressure	Drying Time (h)
Vacuum drying	35	100 mbar	12
Vacuum drying	50	100 mbar	5
Vacuum drying	70	100 mbar	3
Conventional drying	70	atm. *	7
Open sun drying	31.99	atm. *	26

* atm.—atmospheric pressure.

**Table 2 foods-11-00112-t002:** Values of the constants in Peleg’s equation (*K*_1_, *K*_2_) ± S.E., coefficient of determination (*R*^2^) and rate of desorption (*R*_0_) for different drying methods for grape seeds, skins and pomace.

Drying Method	Material	*K*_1_ (h %^−1^) ± S.E.	*K*_2_ (%^−1^) ± S.E.	*R* ^2^	*R*_0_ (% h^−1^)
Vacuum drying (35 °C)	Seeds	7.443 ± 2.868	2.701 ± 0.449	0.953	0.134
	Skins	2.521 ± 0.226	1.023 ± 0.354	0.990	0.397
	Grape pomace	2.166 ± 0.766	1.609 ± 0.120	0.957	0.462
Vacuum drying (50 °C)	Seeds	1.562 ± 0.112	1.924 ± 0.034	0.997	0.640
	Skins	1.099 ± 0.122	0.873 ± 0.043	0.987	0.893
	Grape pomace	1.011 ± 0.133	1.222 ± 0.047	0.992	0.989
Vacuum drying (70 °C)	Seeds	0.644 ± 0.124	2.722 ± 0.079	0.995	1.552
	Skins	0.408 ± 0.064	0.913 ± 0.041	0.989	2.448
	Grape pomace	0.275 ± 0.167	1.482 ± 0.106	0.974	3.634
Conventional dryer (70 °C)	Seeds	0.743 ± 0.238	2.094 ± 0.062	0.995	1.346
	Skins	0.679 ± 0.137	0.937 ± 0.035	0.992	1.473
	Grape pomace	0.419 ± 0.073	1.129 ± 0.019	0.998	2.385
Open sun drying	Seeds	9.705 ± 2.796	2.405 ± 0.183	0.971	1.103
	Skins	6.332 ± 1.046	0.894 ± 0.068	0.971	0.158
	Grape pomace	4.311 ± 0.822	1.192 ± 0.055	0.991	0.242

**Table 3 foods-11-00112-t003:** The results of the activation energy (*E_a_*), coefficient of hydration (*K*_ref_) and coefficient of determination for grape seeds, skins and pomace for the vacuum drying method.

Material	*E_a_* (kJ mol^−1^)	*K*_ref_ (h^−1^)	*R* ^2^
Grape seeds	60.785	32.027	0.956
Grape skins	45.740	59.122	0.999
Grape pomace	52.095	73.729	0.990

**Table 4 foods-11-00112-t004:** Thermodynamic parameters for the drying of grape pomace in a vacuum dryer.

Material	Temperature (°C)	∆*H* (kJ mol^−1^)	∆*S* (kJ mol^−1^ K^−1^)	∆*G* (kJ mol^−1^)
Grape seeds	35	58.223	−0.2844	145.873
	50	58.098	−0.2848	150.143
	70	57.932	−0.2853	155.844
Grape skins	35 50 70	43.178 43.053 42.887	−0.2793 −0.2797 −0.2802	129.258 133.451 139.051
	35	49.553	−0.2775	135.047
Grape pomace	50	49.408	−0.2779	139.213
	70	49.242	−0.2784	144.776

**Table 5 foods-11-00112-t005:** Starting polyphenols, tannins and tartaric acid contents in grape seeds, skins and pomace and percentage of polyphenols, tannins and tartaric acid degradation in grape seeds, skins and pomace during drying under different conditions.

	Material	Polyphenols Content (mg/g)	Tannins Content (mg/g)	Tartaric Acid Content (mg/g)
Initial bioactive compounds contents	Seeds	66.23 ± 3.97	73.88 ± 12.92	0.46 ± 0.12
	Skins	18.67 ± 3.81	29.45 ± 3.84	5.16 ± 0.58
	Grape pomace	29.17 ± 3.54	41.54 ± 8.03	3.54 ± 0.28
**Drying Method**	**Material**	**Polyphenols** **Degradation (%)**	**Tannins** **Degradation (%)**	**Tartaric Acid Degradation (%)**
Vacuum drying (35 °C)	Seeds	n.d.	4.83	n.d.
	Skin	15.34	10.73	9.27
	Grape pomace	n.d.	13.74	14.81
Vacuum drying (50 °C)	Seeds	n.d.	13.74	n.d.
	Skin Grape pomace	n.d. 4.21	1.47 5.38	2.79 n.d.
Vacuum drying (70 °C)	Seeds	n.d.	n.d.	n.d.
	Skin	n.d.	n.d.	5.88
	Grape pomace	n.d.	n.d.	n.d.
Conventional dryer (70 °C)	Seeds	n.d.	32.27	22.81
	Skin	n.d.	31.68	34.83
	Grape pomace	n.d.	22.39	27.01
Open sun drying	Seeds Skin Grape pomace	n.d. n.d. n.d.	3.32 5.27 17.37	11.34 6.27 4.70

n.d.—no degradation.

## Data Availability

The authors confirm that the data supporting the findings of this study are available within the article.

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
