# Peer review of "Comparison of Drying Methods and Their Effect on the Stability of Graševina Grape Pomace Biologically Active Compounds"

_foods, 2022, doi:10.3390/foods11010112_

Round 1

Reviewer 1 Report

The review of the research article "Comparison of drying methods and their effect on the stability of Graševina grape pomace biologically active compounds" was carried out.

The methodology and results are interesting although observations have been made regarding the clarity of its presentation. I believe that there are points to improve to facilitate the future reader a clear understanding. Some inconsistencies were observed regarding concepts on drying fundamentals. The observations were reflected in the form of comments in the attached PDF file.

Reviewer 2 Report

Revision:

Comparison of drying methods and their effect on the stability  of Graševina grape pomace biologically active compounds

The topic is interesting and not explored in scientific journals, thus may gather many readers and brings substantial amount of citations. However the manuscript needs revision in some parts major, before considering publication.

Abstract:

Abstract needs improvement.  The introduction part is too long while description of results is too short terse (not enough informative). For example, the title reflected on biologically active compounds whereas this information are scarce. Please revise and supply appropriate information about obtained results.

Materials and methods:

  1. As I understand the Authors assayed: „the amount of total dry matter and residual moisture 113 of Graševina grape pomace, seeds and skins samples”. Please consider to include some fundamental/basic results in the part Results and Discussion.
  2. Regarding 2.3 part, I suggest incusion scheme or table which show condiotions of all 5 variants drying methods applied in the experiment.
  3. The Authors stored samples until analysis. Please explain how long ? and why samples were stored ?
  4. Linees 201-202: please ensure that tanins and proanthocyanidins are the same compounds.
  5. The description of statistical analysis needs supplementation, which method was used analysis of variance ?

Results and Discussion

  1. In line 281-283, the Authors found „Comparing the rate of desorp-281 tion for seeds, skins and pomace, it can be noticed that the lowest rate was for seeds, then 282 skins and the greater values were obtained for grape pomace.”. Please try to explain why ?
  2. Line 318, I am not sure that the term „energy conservation” is proper. Please verify.
  3. In my opinion part 3.2 needs revision and reformualtion. Why, the obtained results about: total polyphenols content , tanins content and tartaric acid content in fresh and dried samples are not presented (only differences in the contents in Table 4). Please supply some results maybe not for all studient varints od drying, maybe only for the chosen ones.

I found some valuable information about the content in the all three source is incled in Lines 367-369, hoever there is a llack of the data presented for example in the table.

Conlusions

Conclusions needs reformulation. I am not sure that the sentences in lines 431-434 are necessary beacuse not derived from the presented experiment. There is a lack information about tanins which which were more vulnerable on drying conditions as comapred with total polyphenols and tartaric acid.

Keywords;

Please verify there is a lack of studied compounds.

Round 2

Reviewer 2 Report

The major ambiguities of the manuscript have been corrected. I am grateful for answers on all comments. Good luck with your future work.